# Studies on PVP-Based Hydrogel Polymers as Dressing Materials with Prolonged Anticancer Drug Delivery Function

**DOI:** 10.3390/ma16062468

**Published:** 2023-03-20

**Authors:** Agnieszka Sobczak-Kupiec, Sonia Kudłacik-Kramarczyk, Anna Drabczyk, Karolina Cylka, Bozena Tyliszczak

**Affiliations:** 1Department of Materials Engineering, Faculty of Materials Engineering and Physics, Cracow University of Technology, 37 Jana Pawła II Av., 31-864 Krakow, Poland; 2Institute of Inorganic Chemistry and Technology, Faculty of Chemical Engineering and Technology, Cracow University of Technology, 24 Warszawska St., 31-155 Krakow, Poland

**Keywords:** tamoxifen, cyclodextrin, hydrogels, drug delivery systems, sorption properties

## Abstract

Tamoxifen is a well-known active substance with anticancer activity. Currently, many investigations are performed on the development of carriers that provide its effective delivery. Particular attention is directed toward the formation of cyclodextrin–drug complexes to provide prolonged drug delivery. According to our knowledge, carriers in the form of polyvinylpyrrolidone (PVP)/gelatin-based hydrogels incorporated with β-cyclodextrin–tamoxifen complexes and additionally modified with nanogold have not been presented in the literature. In this work, two series of these materials have been synthesized—with tamoxifen and with its complex with β-cyclodextrin. The process of obtaining drug carrier systems consisted of several stages. Firstly, the nanogold suspension was obtained. Next, the hydrogels were prepared via photopolymerization. The size, dispersity and optical properties of nanogold as well as the swelling properties of hydrogels, their behavior in simulated physiological liquids and the impact of these liquids on their chemical structure were verified. The release profiles of tamoxifen from composites were also determined. The developed materials showed swelling capacity, stability in tested environments that did not affect their structure, and the ability to release drugs, while the release process was much more effective in acidic conditions than in alkaline ones. This is a benefit considering their use for anticancer drug delivery, due to the fact that near cancer cells, there is an acidic environment. In the case of the composites containing the drug–β-cyclodextrin complex, a prolonged release process was achieved compared to the drug release from materials with unbound tamoxifen. In terms of the properties and the composition, the developed materials show a great application potential as drug carriers, in particular as carriers of anticancer drugs such as tamoxifen.

## 1. Introduction

Recently, polymer materials are increasingly applied in many areas, including medicine and pharmacy [1,2]. These materials are applied in preparation of wound dressings [3], medical equipment [4], artificial prostheses [5] or organs [6]. Due to the intensive progress in investigations on the properties of polymers, they are also more and more often used in controlled drug delivery systems playing a role of drug carriers [7]. Conventional methods of the delivery of active substances do not fully reflect the therapeutic properties of the delivered pharmacological substances [8]. This is related to the distribution of the drug in the body which begins with its administration that usually takes place via the oral route. As a result, such an active substance is distributed through the whole body. This, in turn, reduces the amount of the dose reaching the desired place. Therefore, many investigations are currently being conducted to develop appropriate polymer carriers that provide target drug biodistribution [9,10].

An important aspect is the targeted delivery of anticancer drugs. Their distribution via conventional routes results usually in the occurrence of serious side effects. An example of such an anticancer drug is tamoxifen, widely applied in breast cancer treatment [11]. For example, Beh et al. proposed nanostructured lipid-based carriers coated with erythropoietin and incorporated with tamoxifen. It was demonstrated that the application of such a combination improved the specificity and safety of applied anticancer substances and showed an anticancer effect against a mammary gland tumor [12]. In another study, Elnaggar et al. reported on the delivery of tamoxifen via drug delivery systems with self-nanoemulsifying properties [13], while Monteagudo et al. developed a Polysorbate-80-based microemulsion for this purpose [14]. Next, Yadav presented studies concerning the application of a drug carrier based on poly(lactic-co-glycolic acid)-poly(ethylene glycol) di-block copolymers and proved that such a solution contributed to drug accumulation near cancer cells [15]. Albert et al. also performed studies on the application of carriers based on poly(d,l-lactice-co-glycolide acid) loaded with magnetite nanoparticles as systems for tamoxifen delivery [16]. Other proposed solution involved the use of drug carriers based on L-α-phosphatidylcholine-based nanostructured lipids [17], chitosan nanoparticles [18], or copolymers of α-tocopherol succinate-g-carboxymethyl chitosan [19]. Another proposed method was the development of a hydrogel system for local drug injection at the tumor site. Thermosensitive hydrogels with tamoxifen were presented, e.g., by Meng et al. and Shaker et al., while both developed compositions provided a prolonged release of tamoxifen at the tumor site [20,21].

Among the various methods of tamoxifen delivery, there have been many attempts to deliver this drug from a carrier in the form of plaster that could be applied externally to the neoplastic site. Thus, the main research subject of this work was to develop a hydrogel dressing based on polyvinylpyrrolidone (PVP) and gelatin-containing gold nanoparticles and cyclodextrin–tamoxifen complexes. Base polymers such as PVP and gelatin are widely applied in medicine and in relative areas, while their adequate treatment allows us to obtain hydrogel material which can also be easily modified already at the stage of the synthesis [22]. Polyvinylpyrrolidone was chosen as the main component of the polymer matrix because of its unique properties. Depending on the molecular weight of the PVP used, the properties of the entire polymer matrix can be controlled. This wide range of capabilities makes it possible to obtain a suitably rigid material. In addition, PVP is considered a polymer that is safe for biomedical applications and exhibits stabilizing effects. Next, gold nanoparticles, which have been of particular interest in anti-cancer therapies in recent years, were also used to obtain composites. Au nanoparticles are biocompatible and can be effectively used in cancer treatment in terms of multimodal therapies, including photodynamic therapy, chemotherapy, radiation therapy and immunotherapy. In addition, due to their ability to be functionalized and have their shape and size controlled, they can also be excellent carriers for active substances. Due to the aforementioned aspects, an attempt was made to obtain composite materials containing gold nanoparticles [23]. In addition, cyclodextrins were also used in the synthesis of hydrogel materials, whose role was to form a stable complex with the active ingredient and prolong drug release capacity [24]. The proposed method of controlled delivery of tamoxifen in a patch has not yet been described in the literature. The advantage of the proposed solution is the possibility of local external application which will reduce the risk of side effects. In addition, the use of a complex of the active ingredient with cyclodextrins is to obtain a carrier with an extended-release profile which is extremely important in this type of system. A dressing with a prolonged release of the active substance will require much less frequent changing, which will also increase patient comfort.

Next, Hu et al. investigated magnetite (Fe_3_O_4_)/poly(L-lactic acid) composite nanoparticles loaded with this active substance [25]. Such materials were prepared using a solvent evaporation/extraction method proceeding in an oil/water emulsion. During the studies, the superparamagnetic properties of the developed carriers were demonstrated. It was also proved that the only approx. 20% of breast cancer cells were viable after four-day incubation with tamoxifen-loaded composites, which confirmed the effectiveness of the developed carriers [26]. Magnetic nanocomposites as tamoxifen carriers were also proposed by Popova et al.; in their work [27], a system consisting of the magnetic iron-oxide-containing MCM-4 silica particles grafted with the chains of poly(ethylene glycol) PEG chains was synthesized and characterized. The research methodology included such analyses as ATR FT-IR spectroscopy, X-ray diffraction, thermogravimetry or N_2_ physisorption, wherein the most important was to verify the tamoxifen release ability via these materials as well as the interactions between such carriers and the active substance. It was demonstrated that the effective release of the drug took place in the environment at pH = 7.0 and, importantly, that developed materials showed antiproliferative properties towards breast cell lines [28].

An interesting solution is the use of cyclodextrins to form a complex of these compounds with active substance. Such a complexation has been presented, for example, by Memisoglu-Bilensoy et al., who synthesized amphiphilic beta-cyclodextrin nanoparticles incorporated with tamoxifen citrate [29]. As a result of the performed experiments, the prolonged release profile of the drug (total release for 6 h) in the case of such a complex was reported, wherein in the case of the release of tamoxifen unbound with cyclodextrin showed a complete and rapid release in just one hour [30]. Studies on such complexes of tamoxifen and cyclodextrins were also conducted by Buchanan et al. [31], Baygi et al. [32], Torne et al. [33], Daoud-Mahammed et al. [34] and Liu et al. [35].

Nonetheless, to our knowledge, the carriers in the form of composites based on polyvinylpyrrolidone (PVP) and containing cyclodextrin–tamoxifen complexes and modified additionally with gold nanoparticles have not been described so far. Importantly, the presented works included both the synthesis of metallic nanoparticles, cyclodextrin–drug complexes, and the obtaining of composites via the photopolymerization process. The first step of performed research works was the preparation of gold nanoparticles while the conditions of the synthesis were also verified by applying a stabilizing–reducing solution with various concentrations. The optical properties of obtained nanomaterials were verified via UV-Vis spectrophotometry, while their size was determined via dynamic light scattering (DLS). The next stage was to synthesize complexes of cyclodextrin with tamoxifen while the kneading method was applied. Finally, polymer composites based on PVP and gelatin and incorporated with tamoxifen or β-cyclodextrin–tamoxifen complexes were obtained. Importantly, two types of composites were prepared, i.e., using tamoxifen and the complex of this active substance with β-cyclodextrin. Then, the composites prepared were characterized, including verifying their behavior in simulated physiological liquids; the impact of such environments on their structure defined via Fourier-transform infrared (FT-IR) spectroscopy was also evaluated. The sorption capacity of composites was then characterized.

## 2. Materials and Methods

### 2.1. Materials

Reagents such as tetrachloroauric (III) acid trihydrate (HAuCl_4_ · 3H_2_O, 99%, d = 3.90 g/mL) and Arabic gum (powder, branched polysaccharide) used for the synthesis of gold nanoparticles were bought from Avantor Performance Materials Poland S.A. (Gliwice, Poland). Polyvinylpyrrolidone (PVP, M.W. 58,000 g/mol) was purchased from Alfa Aesar (Kandel, Germany). 2-hydroxy-2-methylpropiophenone (photoinitiator, 97%, d = 1.077 g/mL), poly(ethylene glycol) diacrylate (crosslinking agent, average molecular weight 700 g/mol, d = 1.12 g/mL), tamoxifen (active substance, ≥99%), gelatin and β-cyclodextrin (CD, powder, ≥97%) were purchased from Sigma Aldrich (Saint Louis, MO, USA).

### 2.2. Synthesis of Gold Nanoparticles

Gold nanoparticles were obtained via the chemical reduction of gold ions derived from tetrachloroauric (III) acid using an adequate stabilizing–reducing agent, i.e., Arabic gum. Firstly, 25 mL of HAuCl_4_ solution containing 500 ppm of Au was introduced into a round-bottom flask and heated to 90 °C under a reflux condenser and with constant stirring. When this temperature was achieved, 25 mL of the Arabic gum solution was put into the reaction mixture and the process was then carried out for 1 h. In order to select the most appropriate conditions of nanogold preparation, various concentrations of stabilizing–reducing agent solution were applied while the size of the particles obtained using various conditions was measured to verify the effectiveness of the reaction performed. The nanogold suspension was stored at ambient temperature and subjected to the investigations aimed at determining the size and optical properties of the prepared particles.

### 2.3. Analysis of the Size of Gold Nanoparticles via Dynamic Light Scattering (DLS) Method

The size of gold nanoparticles obtained was determined using the dynamic light scattering (DLS) technique. Studies were conducted by means of Zetasizer Nano ZS apparatus (Malvern Instruments company, Malvern, UK). Measurements were performed at room temperature.

### 2.4. Analysis of the Optical Properties of Gold Nanoparticles by UV-Vis Spectrophotometry

The optical properties of the gold nanoparticle suspension were characterized using a UV-Vis Evolution 220 spectrophotometer (Thermo Fisher Scientific Inc. company, Waltham, MA, USA). Analyses were performed within the range 400–600 nm with a 1 cm optical length cuvette (a spectral resolution—2 nm) at ambient temperature.

### 2.5. Synthesis of β-Cyclodextrin–Tamoxifen Complexes

Complexes of β-cyclodextrin with a drug were prepared using a kneading method while the mentioned reagents were mixed in a 2:1 molar ratio. For this purpose, β-cyclodextrin was wetted with distilled water (few drops) and kneaded in a ceramic mortar. Then, tamoxifen was added and the whole was kneaded. The paste formed in a mortar was then dried at 37 °C for 24 h. The chemical structure of the obtained powder of the β-cyclodextrin–tamoxifen complex was next characterized via FT-IR spectroscopy.

### 2.6. Preparation of PVP/Gelatin-Based Composites

PVP/gelatin-based composites were prepared via a photopolymerization process. For this purpose, the base solution consisting of an adequate amount of 10% PVP solution and 2% gelatin solution was mixed with the gold nanoparticle suspension, an adequate amount of tamoxifen or the β-cyclodextrin–tamoxifen complex, a crosslinking agent (poly(ethylene glycol) diacrylate) and photoinitiator (2-hydroxy-2-methylpropiophenone). The obtained reaction mixture was mixed carefully, poured into a reaction vessel and treated with UV radiation for 120 s. Compositions of the obtained composite materials are provided below in Table 1.

Figure 1 shows the synthesis scheme and an example of the prepared hydrogel material (sample 1.78 g_complex).

PVP/gelatin-based composites were next subjected to investigations aimed at characterizing their physicochemical properties.

### 2.7. Investigations on Physicochemical Properties of Hydrogels

#### 2.7.1. Analysis of the Chemical Structure of Hydrogels via FT-IR Spectroscopy

Fourier-transform infrared (FT-IR) spectroscopy was applied to verify the presence of functional group characteristics for the tested materials. The study was performed firstly for tamoxifen and β-cyclodextrin, and subsequently for the complex formed from these reagents. Such an analysis was conducted by means of a Nicolet iS5 Thermo Scientific spectrometer, while FT-IR spectra were recorded with a spectral resolution of 4 cm^−1^ within the 4000–400 cm^−1^ range.

#### 2.7.2. Characterization of Swelling Properties of Hydrogel Materials

Analysis of the sorption properties was performed for all prepared hydrogel polymers. For this purpose, samples weighing approx. 1.0 g were prepared from all tested materials. Next, the materials prepared were weighed accurately, and introduced into 50 mL of the tested medium for a selected period of time. The study was carried out in media such as distilled water, Ringer liquid and simulated body fluid (SBF), while the swelling ability of hydrogel samples was verified after 1 h, 24 h and 48 h. After such periods of time, swelled samples were separated from the liquids, excess unbound water was removed and the sample was weighed again. Swelling capability was evaluated via determining the swelling ratio (α), calculated using Equation (1) given below:(1)$$\alpha =\frac{\left(m_{x}-m_{0}\right)}{m_{0}}$$
where α—swelling ratio, g/g; $m_{x}$—mass of sample swelled during a specific period of time (x, while x was 1 h, 24 h and 48 h), g; and $m_{0}$—mass of sample before swelling, g.

#### 2.7.3. Behavior of Composites in Simulated Physiological Liquids

In order to evaluate the behavior of hydrogels in simulated physiological liquids as well as to check whether they degrade in such environments, samples of the materials (with a mass of approx. 1.0 g) obtained were immersed in such liquids as Ringer liquid, SBF and distilled water, i.e., in the same liquids in which swelling ability was verified. The incubation was performed for two weeks at 37 °C and such a temperature was applied to simulate conditions occurring in human body.

After the immersion, samples were dried at 37 °C for 24 h, and two methodologies of drying were applied—samples were dried until their mass stabilized (approx. 30 min for each sample) using a moisture analyzer (1) and in a laboratory incubator at 37 °C for 24 h (2). Next, samples were subjected to FT-IR spectroscopy to verify the impact of such an incubation on their chemical structure and, importantly, their masses were determined and compared to the initial mass (before the incubation). Such studies were performed to verify whether tested samples degraded in environments simulating physiological liquids.

#### 2.7.4. Evaluation of Hydrogels Using an Optical Microscope

The morphology of the hydrogels was evaluated using a Delta Optical Genetic Bino optical microscope. The analysis was performed at ambient temperature.

#### 2.7.5. Characterization of the Mechanical Properties of the Composites

The developed samples were next subjected to studies aimed at determining their mechanical properties, including elasticity and deformation under the tension applied. The study was performed in line with the ISO 527-2 type 5A and ISO 37 type 2 standards. Firstly, a ZCP020 manual blanking press was used to prepare paddle-shape samples. Next, the composite samples were placed between the jaws of the Shimadzu universal testing machine and the analysis was performed until the sample ruptured. Such a procedure allowed us to determine the stress–strain dependance via defining the tensile strength of the tested materials as well as their percentage elongation (A). The tensile strength (R_m_) was calculated using Formula (2) while the percentage elongation (A) was determining using Formula (3), both of which are given bellow.
(2)$$R_{m}=\frac{F_{m}}{S_{0}}$$
(3)$$A=\frac{100\times (I_{u}-I_{0})}{I_{0}}$$
where $F_{m}$—the maximum strength, $S_{0}$—the cross-sectional area of the tested sample before the analysis, $I_{u}$—the measuring length after the sample rupture and $I_{0}$—the measuring length of the sample before the analysis.

#### 2.7.6. Studies on Determining the Release Profile of Tamoxifen

An important aspect of the performed experiments was to determine the release profile of tamoxifen, while the most attention was paid to verify the difference between its release from samples containing tamoxifen as a complex of this drug with β-cyclodextrin and from samples with this pharmaceutic in an unbound form. The possibility of controlled drug release via such dressings increases their application potential and makes them more useful in terms of wound healing processes. Studies were conducted using two media with various pH, i.e., an acidic (pH = 2.0) and an alkaline (pH = 7.4) one. The first conditions were provided by a 2% solution of citric acid while the second one was achieved by phosphate buffer (PBS). Firstly, the composites were introduced into flasks with 200 mL of tested media. Then, the flasks were subjected to the shaking process (via the Hanchen ES-60E Temperature Controlled Incubator and Shaker Scientific Incu-Shaker Shaking Incubator). The study was performed at 36.6 °C (temperature of human organism) and at 80 rpm shaking conditions. After selected time periods, 3 mL of each solution were transferred into the cuvettes and investigated spectrophotometrically to verify the presence of released substances. Importantly, after sampling, the flasks with the tested media were supplemented each time to maintain the same volume of the environment.

#### 2.7.7. Statistical Analysis of the Results of the Investigations

The results of the investigations were subjected to statistical analysis performed via the two-way and one-way analysis of variance (ANOVA) while an alpha value of 5% was applied. All measurements were carried out three times (n = 3) and are presented as an average value with standard deviation (SD).

## 3. Results and Discussion

### 3.1. Analysis of the Stability of the Particles Prepared under Various Stabilizing–Reducing Agent Solution Concentrations

The main task of this part of the investigations was to verify the effectiveness of the concentration of the stabilizing–reducing agent solution in the preparation of stable and monodispersed gold nanoparticles. In Figure 2, the images presenting the color change in the reaction suspension are shown, while the color of the suspension observed directly after adding the stabilizing agent and its color after 60 min of the reaction has been compared.

As a result of all the performed syntheses, a ruby-colored suspension was obtained. In the case of the use of stabilizing–reducing agent solutions at lower concentrations, i.e., 0.5% and 1.0%, the reaction suspension directly after its addition was colorless. After 60 min of the reaction, it became ruby-colored. The use of the stabilizing–reducing agent solutions at higher concentrations, i.e., 2.0% and 3.0%, resulted in the appearance of the ruby suspensions directly after introducing this reagent. These observations allowed us to indicate that in the case of all the applied concentrations of the stabilizer, ruby suspensions of colloidal gold nanoparticles were formed. Nonetheless, it was also important to verify the impact of the stabilizing agent concentration used on the dispersity of the particles. For this purpose, their size was characterized via the DLS technique, and the results obtained are presented below in Figure 3.

Based on the results of the DLS analysis (Table 2), it may be concluded that in the case of all analyzed suspensions, the polydispersity of the particles obtained is observed. However, the lowest polydispersity was observed when 3% Arabic gum solution was used. Moreover, it was concluded that the use of the stabilizing–reducing agent with such a concentration led to the preparation of the highest amount of nanosized particles compared to other variants of the reactions performed. Next, in order to confirm the preparation of gold nanoparticles, and verify their optical properties, UV-Vis spectrophotometric analysis of the suspension obtained using 3% Arabic gum solution as a stabilizing–reducing agent was performed. The UV-Vis spectrum obtained is shown below in Figure 4.

In terms of the phenomenon of the surface plasmon resonance (SPR), the suspension of gold nanoparticles shows an intense ruby color. Considering the occurrence of a characteristic resonance absorption peak resulting from the SPR of gold nanoparticles, it is possible to determine easily their presence in the analyzed suspension using UV-Vis spectrophotometry [33,34]. The absorption band characteristic for gold nanoparticles occurs within the wavelength range 515–570 nm [35]. In the case of the tested suspension, an intense absorption peak with a maximum at a wavelength of approx. 515 nm was observed which clearly indicated the presence of nanogold in the tested medium. This maximum absorption for gold nanoparticles within a similar wavelength range has also been observed in other works [36,37].

### 3.2. Structural Analysis of β-Cyclodextrin–Tamoxifen Complex via FT-IR Spectroscopy

FT-IR spectra of both tamoxifen, β-cyclodextrin, and the complex of these reagents are presented below in Figure 5.

In the case of the FT-IR spectrum of tamoxifen, a characteristic band at approx. 3025 cm^−1^ indicating the stretching vibrations of C–H bond was observed [38]. Next, bands at 1598 cm^−1^ and 1469 cm^−1^, deriving from the stretching vibrations of C–C bond, were identified [39]. A band at 1243 cm^−1^ corresponding to the asymmetric stretching vibrations of the C=C group deriving from the aromatic ring was also observed. Subsequently, a band at 1210 cm^−1^, indicating the occurrence of asymmetric stretching vibrations of phenyl groups was noticed, while a band at 698 cm^−1^, which corresponds to the monosubstituted aromatic benzene, was also reported [40].

In turn, among the absorption bands characteristic for β-cyclodextrin, a wide absorption band of a relatively low intensity within the wavenumber range 3580–3060 cm^−1^, with a maximum intensity at approx. 3315 cm^−1^ which is characteristic of the symmetric and asymmetric vibration of an -OH group, was observed [41]. Next, a band of a low intensity at 1420 cm^−1^, corresponding to the plane-bending vibrations of a hydroxyl group, and adsorption bands at 1369 cm^−1^ and 1333 cm^−1^, indicating the bending vibrations of a -CH group, were identified [42]. Moreover, the band characteristic for the stretching vibrations of –C–O–C- glycosidic bonds was observed at a wavenumber of 1159 cm^−1^ [43]. Next, a low intensity adsorption band with maximum intensities at 1078 cm^−1^ and 1027 cm^−1^ corresponding to the stretching vibrations of C–C and C–O bonds was also noticed [42,43].

In the case of the FT-IR spectrum of CD–tamoxifen complex (black line), the absorption bands corresponding both to the β-cyclodextrin and the active substance (tamoxifen) was identified. Thus, the performed FT-IR analysis confirmed the formation of a complex consisting of both these compounds. Similar conclusions have been drawn during the discussion over the results of experiments aimed at forming a carrier of tamoxifen based on poly(lactic-co-glycolic acid) described in [44].

### 3.3. Analysis of the Chemical Structure of Polymer Composites

Results of the FT-IR analysis of hydrogel materials directly after their synthesis are presented in Figure 6a, while the FT-IR spectra of hydrogels after drying for 24 h are shown in Figure 6b.

In the case of all tested hydrogel materials, a wide absorption band of a high intensity at a wavelength of approx. 3400 cm^−1^, corresponding to a hydroxylic group, was observed. This band is characterized by a high intensity which may indicate a large amount of water in the tested material. Comparing the FT-IR spectra in Figure 6a with the spectra presented in Figure 6b, a significant impact of the 24 h drying process on the chemical structure of the analyzed hydrogels may be noticed. For example, a wide absorption band deriving from –OH group shows a clearly lower intensity for dried samples than for those ones tested directly after the synthesis. Moreover, clear absorption bands, which were not observed previously, may be noticed for samples after the drying process. The FT-IR spectra of all analyzed samples are very similar. Importantly, the absorption bands characteristic of tamoxifen and the CD–tamoxifen complex are difficult to identify due to the fact that the amount of these additives compared to the amount of components forming a polymer matrix is significantly lower. Moreover, functional groups occurring in the structure of tamoxifen and β-cyclodextrin occur also in the structure of the unmodified polymer matrix (black spectrum); therefore, the absorption bands characteristic of these groups may be observed both on the spectra of unmodified samples and hydrogels containing active substances. Thus, the lack of clear differences between the FT-IR spectra of tested samples may result from the overlapping of adsorption bands deriving from the developed polymer materials. Importantly, the absorption bands at 1660 cm^−1^ and 1723 cm^−1^ (marked with orange frame in Figure 6b) may be observed, and they were not previously noticed on the spectra of tamoxifen and β-cyclodextrin. This band is characteristic of the stretching vibrations of a C–O group occurring in the structure of the components of the polymer matrix, i.e., PVP and gelatin [45,46]. Among these two bands, the absorption band at 1660 cm^−1^ shows a higher intensity, which probably results from the fact that this band overlaps with the band characteristic of gold nanoparticles which also occurs within this range [47].

### 3.4. Results of Swelling Investigations

In Figure 7, the results of the investigations on the swelling properties of the hydrogels are presented.

One of the most characteristic features of hydrogel materials are their swelling properties (Table 3). Considering the application of the developed materials as dressings with an active substance release function, a precise characterization of their physicochemical properties is important. Composites have been designed as an external dressing applied, among others, in skin and breast cancer treatment, simultaneously delivering an active substance such as an anticancer drug. During the treatment of the mentioned types of cancer, inflammation as well as the formation of superficial wounds of the skin around the neoplastic lesions is often observed. Thus, the hydrogel dressing showing an ability to absorb the wound exudate and simultaneously release the active substances constitutes an interesting and promising development.

Based on the performed investigations, it may be concluded that all developed materials showed a swelling ability. The highest sorption capacity was identified for samples tested in distilled water. In the case of SBF and Ringer liquid, such an ability has also been observed, but to a lower extent than in the case of distilled water. It was also concluded that the sorption properties depend strongly on the type of the liquid absorbed. Distilled water does not contain any ions, and as a result, this liquid penetrates easily into the polymer network which results in the swelling of a hydrogel sample immersed in this medium. Other tested liquids, i.e., SBF and Ringer liquid, consist of various mono- and divalent ions and therefore the penetration of these liquids into the analyzed samples is limited in terms of possible interactions between the sample and the tested medium. Divalent ions may affect the formation of additional crosslinks within the polymer matrix, simultaneously increasing its crosslinking density and thus limiting the amount of free spaces available for absorbed liquid. Such a phenomenon has not been observed in the case of distilled water; therefore, the highest values of swelling ratios were calculated exactly for this medium.

Apart from the type of the medium applied for swelling, the composition of the tested sample also has a significant impact on its swelling properties. For example, it may be observed that unmodified samples (without active substances) showed the lowest swelling ratios. Next, a slightly higher sorption capacity was observed for samples containing tamoxifen, i.e., 0.25 g_Tamoxifen and 0.50 g_Tamoxifen samples. Importantly, in the case of these samples, an increase in their mass was the most noticeable in the first hour of swelling, while in the next course of the study, their swelling ratios stabilized and further changes in their values were slight. Other behavior was observed in the case of samples containing the CD–tamoxifen complex—here, an increase in the swelling ratios compared to their values calculated for unmodified hydrogels was observed but these ratios increased constantly over the course of the entire study. For example, in the case of the swelling of sample 0.25 g_Tamoxifen in distilled water for 1 h, its swelling ratio was 2.44 g/g, while after 48 h, its value increased by 0.14 g/g. Next, when analyzing the sorption capacity of samples containing only tamoxifen and containing a complex of this drug with β-cyclodextrin (sample 1.78 g_complex), hydrogels with the mentioned complex showed twice the change in the swelling ratio (an increase by 0.29 g/g) during the same swelling period. Based on the results obtained, it may be supposed that in the case of samples containing only tamoxifen (not its complex with β-cyclodextrin), this drug was probably rapidly released from the swelled material during the first hour of the study. The active substance occupying previously free spaces between polymer chains is replaced by absorbed liquid, and therefore a significant increase in the swelling ratio of such a sample was observed. Next, a further increase in its value was not noticed, which may be the evidence of the release of the total amount of tamoxifen in the first hour of the study, after which the tested system stabilized and its sorption was constant. Another situation may be observed in the case of samples containing a CD-drug complex. Here, the swelling ratio of the tested samples after 1 h is lower than in the case of a sample modified with tamoxifen, and its value increased throughout the entire study. After 24 h and 48 h of swelling, step changes in the values of swelling ratios were still observed. Thus, it may be concluded that tamoxifen introduced into the polymer matrix in the form of a cyclodextrin-related drug is released more slowly and, importantly, gradually, while spaces occupied previously by these complexes are filled with absorbed liquid. In terms of the potential applications of the developed materials, the fact that they are able to provide a prolonged active substance release profile is undoubtedly their great advantage. Moreover, the results obtained are consistent with the results of other investigations, in which it was proved that the application of an adequate carrier for tamoxifen provides its prolonged release into the affected site [48,49,50].

### 3.5. Results of the Determination of the Behavior of Composites in Simulated Physiological Liquids

The pH values of simulated physiological liquids in which the composites were incubated are presented below in Figure 8.

Based on the results of performed investigations, it may be concluded that in the case of hydrogel materials containing tamoxifen (i.e., sample 0.25 g_Tamoxifen and 0.50 g_Tamoxifen), a slight increase in the pH of the incubation media at the beginning of the study may be observed. Such a change was probably related to the release of this active substance from the polymer matrix. In Section 3.4 of this paper, it was demonstrated that the tested materials showed swelling properties. The swelling of the hydrogels includes the penetration of water into the polymer network which, in turn, occurs with the simultaneous release of the active substance located between polymer chains. Due to the fact that tamoxifen forms an alkaline solution, the release of this compound into the incubation media thus increases its pH value (Table 4).

In the case of the incubation of samples containing the complex of tamoxifen with β–cyclodextrin, a pH decrease in the incubation media at the beginning of the research was reported. This was also related to the release of this complex into the incubation media. Such a change is due to the fact that β–cyclodextrin forms an acidic solution. However, during further a incubation period, the pH of the incubation media slightly increased which, in turn, is due to the release of tamoxifen from the previously released β–cyclodextrin–tamoxifen complex.

The mentioned changes in the pH of the incubation media were the most visible in the case of distilled water. Other tested liquids, SBF and Ringer liquid, contain various ions which may affect the crosslinking density of the hydrogels. This, in turn, limits the penetration of water into the polymers, and as a result, limits at the same time the release of the active substance from the tested materials. Thus, the pH changes in the case of the incubation of samples in the mentioned liquids were slightly smaller than in the case of distilled water.

In order to verify precisely the impact of the 10-day incubation of hydrogels in simulated physiological liquids on their chemical structure, samples after incubation were subjected to the FT-IR spectroscopy. The FT-IR spectra obtained are presented below in Figure 9.

Based on the results of FT-IR spectroscopy, it may be concluded that the FT-IR spectra of samples before incubation were significantly similar. The only differences were the lower intensities of some absorption bands on the spectra of samples after incubation. These differences were very slight and were probably due to the release of active substances from polymer matrices. The slight pH changes observed during the incubation of the hydrogels may also confirm the release of these substances. Thus, the bands corresponding to the functional groups present in the structure of these additives became slightly less intense. Nonetheless, in the case of all incubated samples, these changes were slight and, importantly, the same bands may be observed both in the case of samples before and after incubation (no disappearance of any of them, which might indicate a degradation of the tested samples).

Additionally, below, in Table 5, weights of the samples measured during the two-week incubation in simulated physiological liquids are shown.

### 3.6. Characterization of Hydrogels Using Optical Microscopy

The images of prepared hydrogel samples are shown in Figure 10.

The microscopical analysis performed allowed us to characterize the morphology of the hydrogels obtained. It may be observed that the morphology of the unmodified sample (Figure 10a) is clearly different from the morphology of samples containing tamoxifen or the β-cyclodextrin–drug complex. The unmodified hydrogel showed a homogeneous surface. In the case of samples containing drugs, their surface is heterogeneous. On the images of hydrogels 0.25 g_Tamoxifen (Figure 10b) and 0.50 g_Tamoxifen (Figure 10c), the particles of the introduced drug (tamoxifen) may be observed. They are, in turn, not visible in the case of images showing samples 1.78 g_complex (Figure 10d) and 3.56 g_complex (Figure 10e)—here the drug was introduced into the polymer matrix in the form of its complex with β-cyclodextrin. This confirmed that the form of the drug applied also affects the morphology of the samples obtained.

### 3.7. Evaluation of the Mechanical Properties of the Composites

Below, in Figure 11, the results of the mechanical measurements such as the tensile strength and the percentage elongation of tested composites are presented.

Based on the results obtained, it may be concluded that the composition of the tested composites strongly affects their mechanical properties (Table 6). In the case of the elongation measurements, it was observed that the highest percentage elongation (23%) was reported for the unmodified sample. In turn, composites containing tamoxifen showed the elongations of 18.0% and 16.5%, while samples containing the β-cyclodextrin–tamoxifen complex exhibited the value of this parameter at the level of 13.0% and 11.0%, respectively. On the other hand, composites with the β-cyclodextrin–drug complex showed, at the same time, the highest tensile strength.

In the case of all the measured parameters, the modification of the samples with both tamoxifen unbound from the complex with β-cyclodextrin and with the β-cyclodextrin–tamoxifen complex affected their elongation and the tensile strength. Samples containing the unbound active substance showed higher tensile strength than unmodified material (sample “0”). This results from the chemical structure of tamoxifen which contains three aromatic rings. Thus, the introduction of such a structure between the polymer chains limit their mobility and, thereby, a more compact and rigid structure is formed compared to the structure of the materials without this additive. In turn, a very complex structure of β-cyclodextrin and its interaction with the polymer chains is reflected in the highest tensile strength of the composite containing the β-cyclodextrin–tamoxifen complex and at the lowest elongation of this sample.

### 3.8. Investigations Determining the Release Profile of Tamoxifen from Developed Composites

In Figure 12, the results of studies on the release profile of tamoxifen from the developed composites in two environments., i.e., an acidic and an alkaline one, are presented.

Based on the performed investigations, it was proven that the developed composites showed an ability to release an active substance from their interior. Nonetheless, it may be observed that the release process was much more effective in an acidic environment. Gelatin which constitutes a component of the composite matrix contains amino groups on long branchings of the main polymer chain. Such groups undergo a protonation process in acidic conditions and NH_3_^+^ ions are formed. These ions, in turn, repel each other electrostatically. As a result of such interactions, the spaces between polymer chains become greater and the active substance may be released more easily than in the case when polymer chains are tightly packed. In the structure of the gelatin, carboxylic groups also occur, which may dissociate and form COO^−^ ions. These may also repel each other electrostatically, but such structures are placed on short branchings of the main polymer chain. As a result, these interactions are of less importance then those ones between amino groups. This is important in terms of the potential application of developed composites, i.e., as carriers of cytostatics. The environment near cancer cells shows an acidic pH (due to the products of cancer cell metabolism); thus, a much more effective release in such conditions than in alkaline ones makes developed carriers promising in terms of their potential application.

Importantly, it was demonstrated that the release of tamoxifen when this active substance is in the composite matrix as a complex with β-cyclodextrin proceeded in a prolonged way. After 120 min, the total amount of tamoxifen was released from the composites containing this drug unbound in the complex, while in the case of the second type of composites (incorporated with complex), a total release was observed after 180 min. This, in turn, demonstrated the fact that the application of β-cyclodextrin as a drug complexing factor contributed to the preparation of carriers with a prolonged drug delivery function. In such a form, tamoxifen was released both from the complex and the composite, resulting in its prolonged delivery to the tested media.

The results obtained are consistent with work previously presented. Analogous results were also presented by Mircioiu et al., who showed that tamoxifen release occurs more efficiently with simulated gastric fluid with a pH = 1.2 than with simulated intestinal fluid. In addition, they also indicated a similar release time, as they proved that 60% of the active substance is released in about two hours [51]. Moreover, similar results were also reported by Taupitz et al. and Yadav et al. [52,53].

## 4. Conclusions

The concentration of Arabic gum strongly affected the stability and dispersity of gold nanoparticle suspensions obtained using this stabilizing–reducing agent. The highest stability and, simultaneously, the lowest dispersity of gold nanoparticles were achieved using a 3% Arabic gum solution.

The preparation of the gold nanoparticle suspension was confirmed via UV-Vis spectrophotometry—an intense absorption peak with a maximum at a wavelength of approx. 515 nm, which is characteristic of the mentioned colloidal metallic nanoparticles, was observed on the obtained UV-Vis spectrum.

On the obtained FT-IR spectrum of the CD–tamoxifen complex, absorption bands characteristic both of β-cyclodextrin and tamoxifen were observed; thus, it may be evidence of the effectiveness of the kneading method applied for the preparation of β-cyclodextrin–drug complex acting as a carrier of an active substance.

FT-IR analysis of the developed materials allowed us to identify the functional groups characteristic for both components forming a polymer matrix as well as for the introduced additives, while a significantly higher intensity was reported for samples subjected to the 24 h drying.

All the developed materials showed swelling properties, while samples containing tamoxifen were characterized by a higher sorption capacity compared to unmodified hydrogels. This was probably due to the release of this active substance from the tested materials which, in turn, led to the formation of free spaces between polymer chains available for the absorbed liquids.

It was shown that the swelling properties of the materials containing the CD–tamoxifen complex increased with time, which may indicate a gradual and prolonged release of the active substance when it was added in the form of a complex with β-cyclodextrin. Another situation took place in the case of samples containing tamoxifen instead of its complex, when such a release occurred rapidly during the first hour of the study.

The FT-IR analysis of samples after 10 days of incubation in simulated physiological liquids showed no changes in the structure of the tested materials, which may have indicated their degradation.

It was proven that the developed composites showed an ability to release the active substance. The release process was much more effective in an acidic environment, which is promising in terms of their potential application as carriers of cytostatics. This is due to the fact that the environment near cancer cells has an acidic pH. Importantly, the introduction of the active substance (tamoxifen) in the form of its complex with β-cyclodextrin resulted in a prolonged release process.

Considering features of the developed hydrogel materials such as high sorption capacity, stability in simulated physiological liquids and an ability to release of active substance from their interior (rapid or prolonged depending on the fact of whether it is applied as a complex with β-cyclodextrin or not), it may be supposed that these materials show a great application potential for application as dressing materials with a controlled drug delivery function and they should be subjected to more advanced studies including determining the release profile of the active substance as well as characterizing their cytotoxicity towards selected cell lines.

## Figures and Tables

**Figure 1 materials-16-02468-f001:**
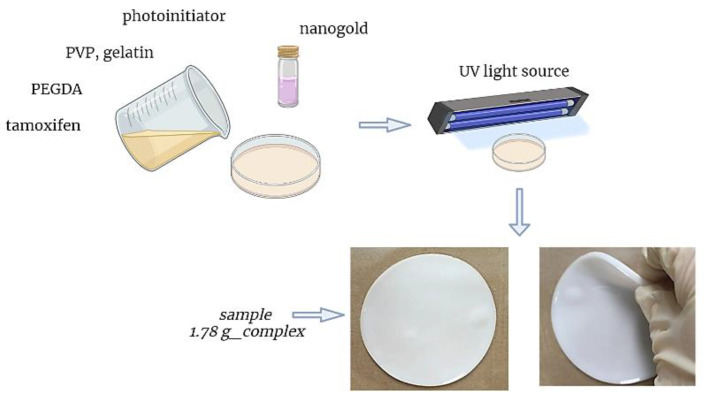
Scheme of synthesis of hydrogel materials.

**Figure 2 materials-16-02468-f002:**
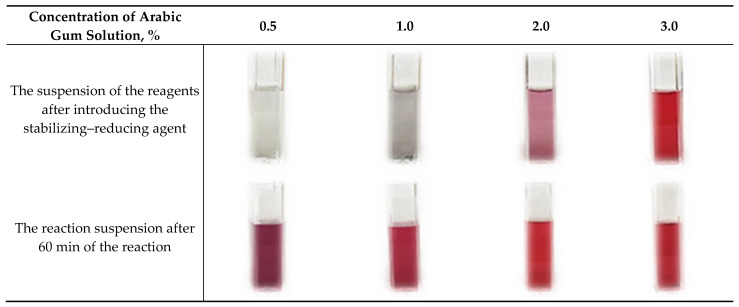
Color changes in the reaction suspensions during the synthesis of nanogold performed using Arabic gum solutions of various concentrations.

**Figure 3 materials-16-02468-f003:**
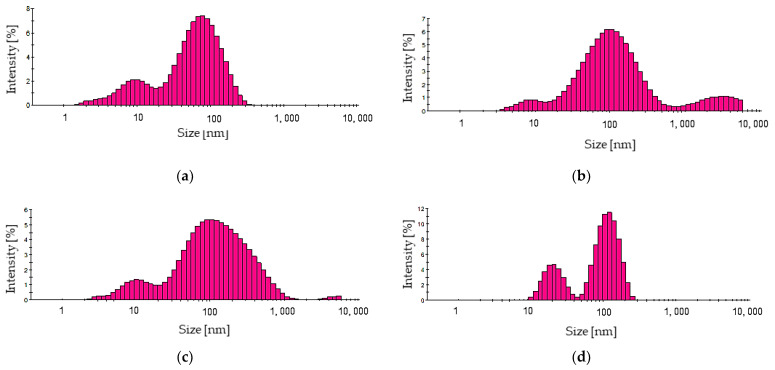
The size distribution of gold particles obtained using 25 mL of 0.5% (**a**), 1.0% (**b**), 2.0% (**c**) and 3% (**d**) Arabic gum solution.

**Figure 4 materials-16-02468-f004:**
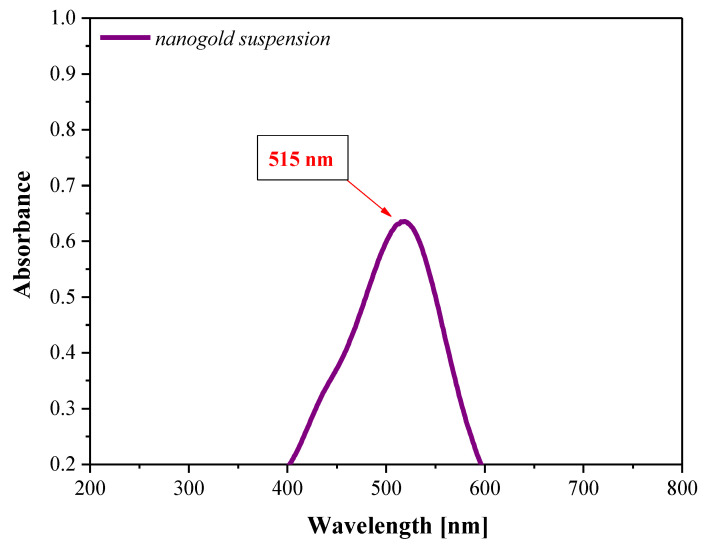
UV-Vis spectrum of nanogold suspension obtained using 3% Arabic gum as a stabilizing–reducing agent.

**Figure 5 materials-16-02468-f005:**
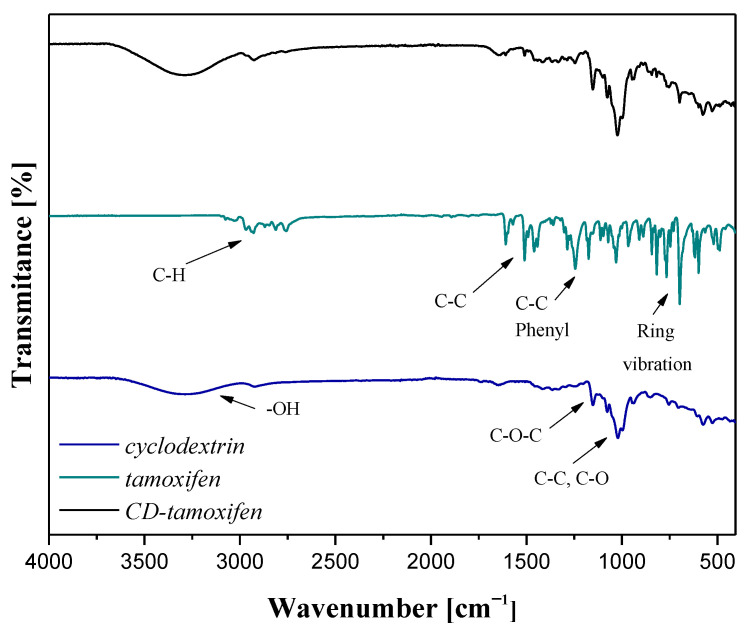
FT-IR spectra of tamoxifen, β-cyclodextrin and their complex (in a molar ratio 2:1).

**Figure 6 materials-16-02468-f006:**
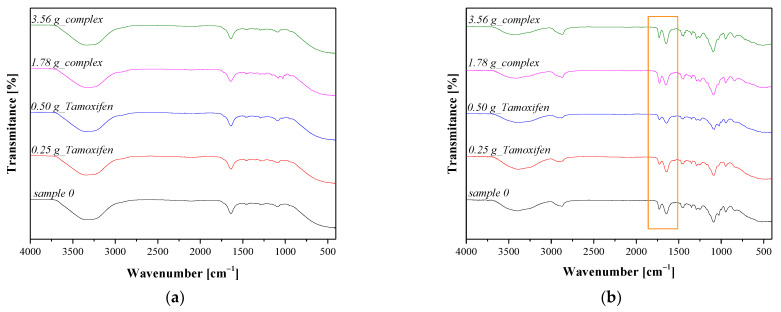
FT-IR spectra of hydrogels directly after their synthesis (**a**) and after 24 h of drying (**b**).

**Figure 7 materials-16-02468-f007:**
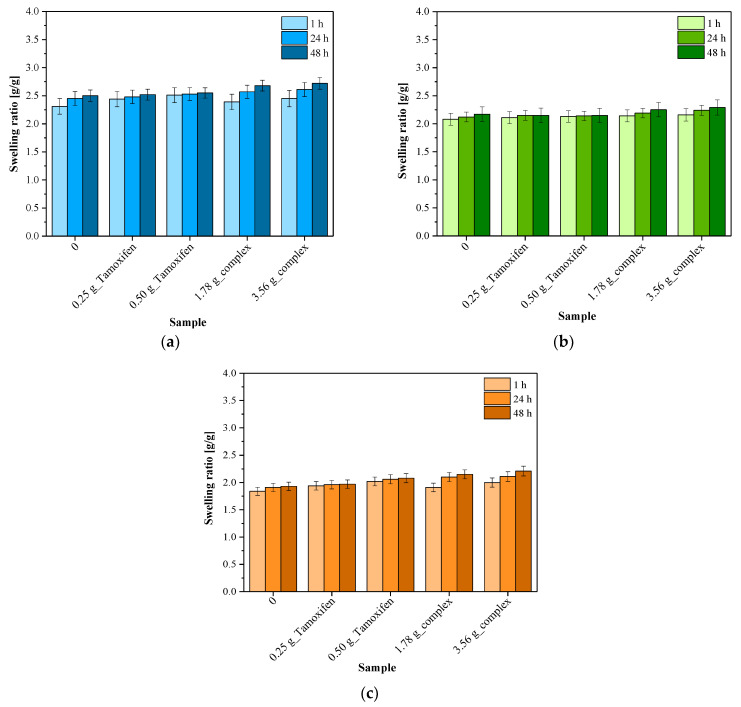
Results of swelling studies of hydrogels in distilled water (**a**), Ringer liquid (**b**) and SBF (**c**) (n—number of repetitions, n = 3).

**Figure 8 materials-16-02468-f008:**
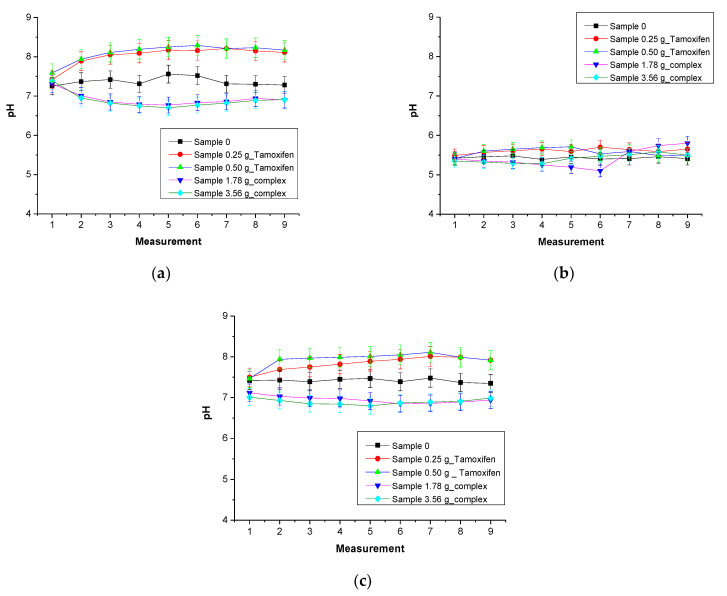
Results of incubation studies in distilled water (**a**), Ringer liquid (**b**) and SBF solution (**c**) (n—number of repetitions, n = 3).

**Figure 9 materials-16-02468-f009:**
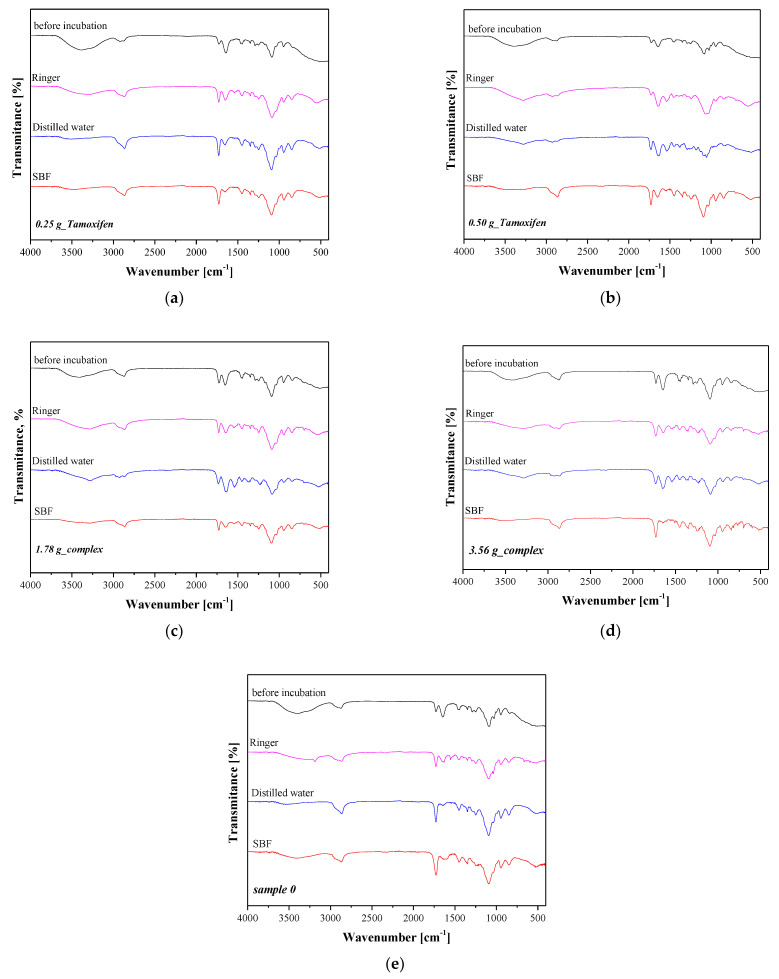
FT-IR spectra of hydrogel samples: 0.25 g_Tamoxifen (**a**), 0.50 g_Tamoxifen (**b**), 1.78 g_complex (**c**) and 1.78 g_complex (**d**) and 3.56 g_complex (**e**) 0.0 g_Tamoxifen.

**Figure 10 materials-16-02468-f010:**
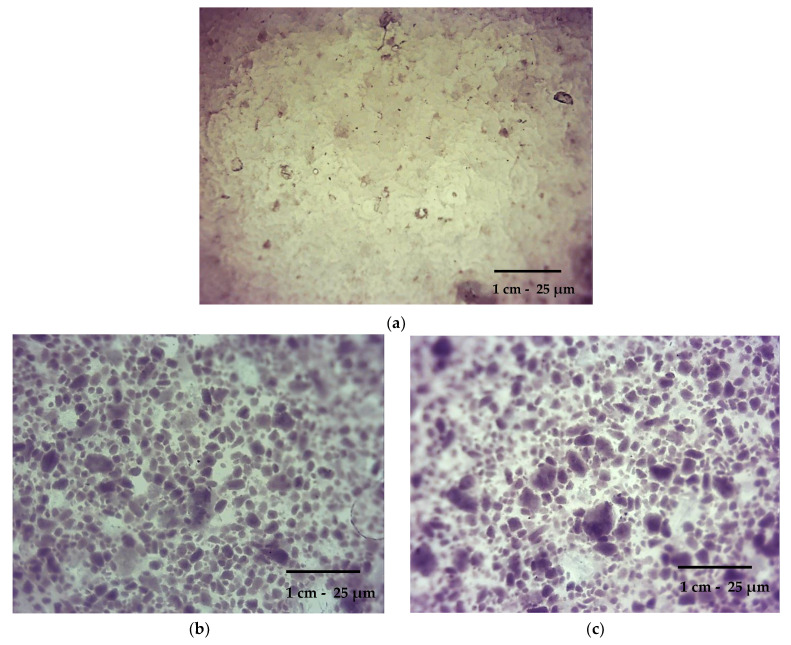
Images of samples: 0 (**a**); 0.25 g_Tamoxifen (**b**); 0.50 g_Tamoxifen (**c**); 1.78 g_complex (**d**); and 3.56 g_complex (**e**).

**Figure 11 materials-16-02468-f011:**
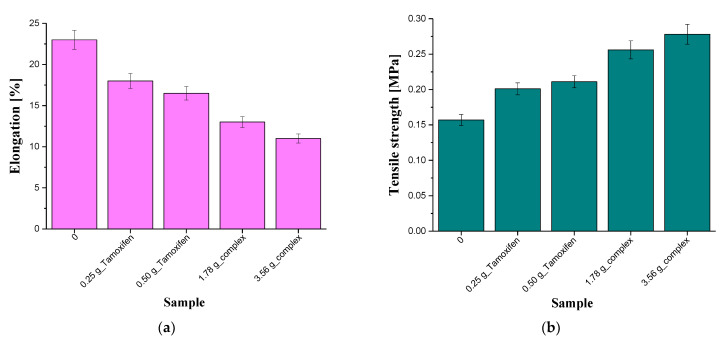
Results of the mechanical studies of the composites showing their elongation (**a**) and the tensile strength (**b**) (n—number of repetitions, n = 3).

**Figure 12 materials-16-02468-f012:**
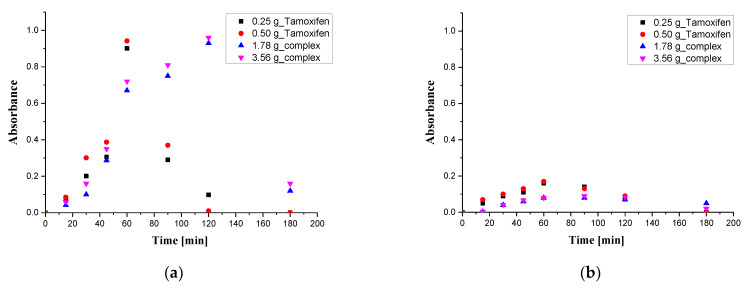
Results of release studies of tamoxifen from the composites in an acidic environment (**a**) and alkaline environment (**b**).

**Table 1 materials-16-02468-t001:** Compositions of polymer composites.

Sample Name	Tamoxifen, g	CD- Tamoxifen, g	Base Solution *, mL	Cross-Linking Agent, mL	Photoinitiator, mL	Nanogold Suspension, mL
0	-	-	10 *	1.60	0.25	1
0.25 g_Tamoxifen	0.25	-				
0.50 g_Tamoxifen	0.50	-				
1.78 g_complex	-	1.78				
3.56 g_complex	-	3.56				

* base solution (8 mL of 10% PVP + 2 mL of 2% gelatin).

**Table 2 materials-16-02468-t002:** PDI coefficient for nanoparticles obtained using different concentrations of gum arabic.

Concentration of Arabic Gum, %	PDI
0.5	0.641
1.0	0.663
2.0	0.874
3.0	0.198

**Table 3 materials-16-02468-t003:** Statistical analysis of obtained data based on the two-way analysis of variance (ANOVA) with repetitions.

Independent Variable	Sum of Squares	Mean Square	f-Value	*p*-Value
Type of incubation fluid	1.73016	0.86508	3414.79825	1.01211 × 10^−21^
Composition of samples *	0.09619	0.02405	94.92105	1.11022 × 10^−16^
Interaction	0.03128	0.00391	15.43421	1.03399 × 10^−8^

* At the 0.05 level, the population means of “type of incubation fluid” are significantly different. At the 0.05 level, the population means of “composition of sample” are significantly different. At the 0.05 level, the interaction between both factors is significant.

**Table 4 materials-16-02468-t004:** Statistical analysis of obtained data based on the two-way analysis of variance (ANOVA) with repetitions.

Independent Variable	Sum of Squares	Mean Square	f-Value	*p*-Value
Type of incubation fluid	48,094.71142	24,047.35571	3.7862	0.03418
Composition of samples *	101,440.34904	25,360.08726	3.99289	0.0103
Interaction	203,387.82154	25,423.47769	4.00287	0.00249

* At the 0.05 level, the population means of “type of incubation fluid” are significantly different. At the 0.05 level, the population means of “composition of sample” are significantly different. At the 0.05 level, the interaction between both factors is significant.

**Table 5 materials-16-02468-t005:** Masses of samples verified during their incubation in physiological liquids.

	**Analytical Balance, Weight [g]**				
	**Sample**				
Degradation liquid	sample_0	0.25 g_ tamoxifen	0.50 g_ tamoxifen	1.78 g_ complex	3.56 g_ complex
Ringer liquid	0.893	0.887	0.887	0.856	0.887
SBF	0.897	0.895	0.885	0.878	0.859
Distilled water	0.889	0.887	0.868	0.865	0.857
Phosphate buffer	0.895	0.879	0.892	0.855	0.858
Citric acid solution	0.883	0.886	0.878	0.875	0.854
	**Moisture analyzer, weight [g]**				
	**Sample**				
Degradation liquid	sample_0	0.25 g_ tamoxifen	0.50 g_ tamoxifen	1.78 g_ complex	3.56 g_ complex
Ringer liquid	0.895	0.885	0.887	0.857	0.887
SBF	0.894	0.894	0.885	0.878	0.857
Distilled water	0.887	0.883	0.869	0.868	0.855
Phosphate buffer	0.896	0.875	0.895	0.857	0.854

**Table 6 materials-16-02468-t006:** Statistical analysis of obtained data based on the one-way analysis of variance (ANOVA) with repetitions.

Independent Variable	Sum of Squares	Mean Square	f-Value	*p*-Value
Composition of samples (elongation) *	300.56667	75.14167	128.81429	1.47688 × 10^−8^
Composition of samples (tensile strength) *	0.02772	0.00693	3248.90625	1.66533 × 10^−15^

* At the 0.05 level, the population means of “composition of sample” are significantly different for both elongation and tensile strength.

## Data Availability

The data presented in this study are available on request from the corresponding authors.

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
