# Peer review of "Studies on PVP-Based Hydrogel Polymers as Dressing Materials with Prolonged Anticancer Drug Delivery Function"

_materials, 2023, doi:10.3390/ma16062468_

Round 1

Reviewer 1 Report

The manuscript "Studies on PVP-based hydrogel polymers as dressing materials with prolonged anticancer drug delivery function" is an interesting and novel approach to incorporate PVP, a biocompatible polymer together with cyclodextrin and AuNP into hydrogels. The hydrogels were tested by different methods.

Following comments need to be taken care of:

1-Figure 3: c and d are marked above the line. A minor change to improve the figures is needed.

2-Figure 4: improve the description on the x axis according to journal. It must be [nm]. It would be better to demonstrate on the same figure all the three concentrations. Even PVP and tamoxifen, and cyclodextrin in free form could be added, to see, how the changes happen. Without these, the figure 3 is very basic and does not allow any other discussions.

3-Figure 5: needs [cm-1] in this form for the x-axis. Otherwise, recommended is to add the FTIR of PVP also and discuss it together with the other results. 

4-PVP and its role are not mentioned specifically. Both Figure 4 and 5 could also explain the role of PVP within the complex. Therefore, adding PVP into both figures will help. How is the release pattern related to PVP?

5-all of the other figures need to use [ ] brackets

6-The references section needs to be edited again

Reviewer 2 Report

This manuscript describes the development of a PVP/Gelatin/gold nanoparticle-based dressing material for prolonged delivery of β-cyclodextrin-tamoxifen complexes. The manuscript characterized the properties of gold nanoparticles, hydrogels, as well as drug release from the material. The manuscript shows the potential of this material for clinical applications. Please see the following questions and comments for modification of the manuscript.

1. Figure legend for most of the figures is not informative enough. A good legend should explain the figure clearly and thoroughly, providing readers with all the information necessary to understand the figure without returning to the main text.

2. Number of samples should be indicated in figures 8, 9, 10, 11, 12, 13, 18, 19, 20. Statistical analysis should also be applied.

3. The manuscript does not give a reason for why gold nanoparticle was used in this material. What is the effect of gold nanoparticles on the material?

4. Line 289, the author said “lowest polydispersity was observed when 3% arabic gum solution was used.” Please provide PDI value comparison of all the samples.

5. Figures 14-16 can be combined into one figure.

6. In table 2, what is the unit for the mass?

7. In Figure 19 and Figure 20, the absorbance shows an increase in the 1st hour and then reduced to 0 at 2 or 3 hours (depending on the sample). However, based on the description in 2.7.7, 3 mL of solution was transferred out and 3 mL of solution was replenished into a total of 200 mL tested media, I would expect the measurement to plateau instead of going down to 0 after a few rounds of sampling. Please explain this phenomenon. 

Reviewer 3 Report

The authors Sobczak-Kupiec et al. Described a new drug delivery system for prolonged release of anticancer drugs. some points might need to be discussed more in detail.

(1)   For the introduction, it may make sense to briefly describe what is the ideal therapeutic goal for prolonged/sustained release. What are the advantages of the authors' proposed system in terms of route of administration, interval, dosage and adverse effects, if compared to the most state-of-the-art treatments.

(2)   The authors used PVP as a substrate to prepare the drug delivery system and PVP is not biodegradable. What is the designed extinction pathway of this drug delivery system after drug release.

(3)   Section 2.7.7: The pH of tumor tissue can vary depending on different factors, such as the type of tumor, the stage of metastasis, and the surrounding environment. However, the most known pH range is between 6.5 and 7.4. The authors tested the release system in a dissolution medium with a pH of 2. Is this physiologically meaningful?

(4)   Section 2.7.7 the study was performed at 36.6 °C. Could the authors explain why this temperature was chosen not 37 °C or higher?

Reviewer 4 Report

The author has conducted a study on hydrogel carriers that can effectively deliver the drug tamoxifen, a well-known anti-cancer drug for breast cancer. Drug release properties are very important in drug carriers and for this purpose, the author has performed various property analyses on the tamoxifen complex.

Also, nanogold has been used in this complex, but I am wondering what is the main role of nanogold in the various analysis results.

The size of the hydrogel-tamoxifen complex made is large, is this complex designed for implantation?

Please provide a brief comparison of the drug release capability of the tamoxifen complexes created by the authors compared to other similar studies.

minor

1. please organize figures from the same experimental group into one figure (there are too many figures overall)

2. please review English spelling

3. please correct the numbering of references.

4. increase the resolution and style of the inserted figures.

5. insert a scale bar in Figure 17.

Reviewer 5 Report

In this manuscript, the authors synthesized PVP-based hydrogel polymers for the delivery of tamoxifen. They characterized the hydrogels with FT-IR, swelling, etc. However, the manuscript was not well organized, and it isn't easy to read and understand. The detailed comments were listed as follows:

1)     The hydrogel preparation method should be continuous, then the analysis and characterization. Please modify the corresponding part.

2)     I strongly suggest the authors add a scheme of the hydrogel synthesis in the revised manuscript, which will be friendly for the readers.

3)     There are too many figures in the manuscript. Please combine or remove some into the supplementary data. 

4) Figures 2, 4, 7, 11, 12, and table 2 could be removed to supplementary data. 

5)     Combine figures 8, 9, and 10; 14, 15, and 16; 19, and 20 into a single figure, respectively.

6)   Figure 4, please add the adsorption beyond 0.2.

7)     The English expression should be improved, there are too many spelling errors and mistakes.

Round 2

Reviewer 1 Report

Well done.

Congratulations

Author Response

Thank you very much.

Bozena Tyliszczak

Reviewer 2 Report

The authors have addressed most of the issues. 

Minor: in table 3, technically, the p-value cannot equal 0. 

Author Response

Dear Reviewer,
Thank you very much for pointing out this mistake. The statistical calculation in the program indicated a result of E-20, but the calculation result is E-21 therefore the calculation program gave a result of 0. I have corrected this error. The result in the table is 1.01211E-21.
Yours sincerely 

Reviewer 4 Report

All issues have been properly addressed.

Author Response

Thank you very much.

Bozena Tyliszczak

Reviewer 5 Report

I have reviewed the revised version of the manuscript, and most of the issues have been revised. Thus I recommend ACCEPT in the present form.

Author Response

Thank you very much.

Bozena Tyliszczak